# The Biosurfactants Mannosylerythritol Lipids (MELs) as Stimulant on the Germination of *Lactuca sativa* L.

**Renato Dias Matosinhos** [1,†], **Karina Cesca** [2], **Bruno Augusto Mattar Carciofi** [2], **Débora de Oliveira** [1,2] and **Cristiano José de Andrade** [1,3,*]

1. Graduate Chemical Engineering Program (PósENQ), University of Santa Catarina (UFSC), Florianópolis 88040-900, SC, Brazil; rd.matosinhos@gmail.com (R.D.M.); debora.oliveira@ufsc.br (D.d.O.)
2. Graduate Food Engineering Program (PPGEAL), University of Santa Catarina (UFSC), Florianópolis 88040-900, SC, Brazil; karinacesca@gmail.com (K.C.); bruno.carciofi@ufsc.br (B.A.M.C.)
3. Department of Chemical Engineering and Food Engineering, Federal University of Santa Catarina, Florianópolis 88040-900, SC, Brazil
* Correspondence: eng.crisja@gmail.com
† This paper is a part of the Master Dissertation of Renato D. Matosinhos, presented at Federal University of Santa Catarina (Brazil).

**Abstract:** The application of pesticides in agriculture leads to improved crop quality and promotes high productivity. However, the uninterrupted use of these chemicals is directly related to environmental impacts, affecting biodiversity and the health of ecosystems and humans. In this sense, mannosylerythritol lipids (MELs) are a promising alternative, as they are biosurfactants with antimicrobial, amphiphilic characteristics, and low toxicity. Thus, in search of a partial reduction in the use of chemical pesticides in agriculture, this work aimed to evaluate the biostimulant effect of one of the homologs of MELs–MEL-B on the germination of Monica lettuce seeds (*Lactuca sativa* L.) and the influence on plant growth and root development. The seeds germinated in different concentrations of MEL-B. The incidence of germinated seeds, the germination index, and the average germination time were evaluated. MEL-B at 158 mg/L stimulated seed germination, growth, and seedling development parameters by 65%, while concentrations of 316 and 632 mg/L did not exceed 45% for these parameters. It was observed that MEL-B at 158 mg/L biostimulated the appearance of lateral roots and promoted only 7% of root stress, a difference of 47% for roots grown with MEL-B at 632 mg/L. Furthermore, MEL-B at 158 mg/L was the highest concentration at which there was no phytotoxic effect of MEL-B on seeds. The increase in enzymatic activity corroborates the phytotoxic effect and seed stress at concentrations of 316 and 632 mg/L, showing results of 47% and 54% of stressed roots. In an unprecedented way, this study proved that MEL-B has a biostimulant and phytotoxic effect related to its concentration.

**Keywords:** agriculture; enzyme activity; stressed roots; glycolipid; lettuce

## 1. Introduction

The application of pesticides in agriculture leads to higher productivity. However, pesticides are easily diffused in soil, air, and water—resulting in large environmental impact [1,2]. In addition, they are vectors for simple and chronic human health problems, such as nausea and headaches, diabetes, and cancer [3]. The commercialization of active ingredients used in the production of pesticides exceeds 4 million tons annually [4]. In Brazil, since 1990, the use of pesticides has been increasing over time. In 2019, approximately 13.300 chemicals were registered [5]. Pesticides are essential to high crop yields and a high level of quality. On the other side, modern agriculture should be efficient and also environmentally friendly.

Biostimulants are eco-friendly compounds that can stimulate plant metabolism and improve the absorption of nutrients in the soil [6]. Biostimulants can be classified into





four main groups based on amino acids and protein hydrolysates; humic substances; microorganisms; and inoculum and algae extracts [7].

To date, the biostimulant potential of biosurfactants has been subtly investigated. For example, the inhibitory effects (in vitro) of *Pseudozyma aphidis* metabolites on phytopathogenic fungi were studied. The results indicate that *P. aphidis* has a potential application as a biocontrol agent for fungal pathogens [8]. In another study, the authors applied *Rhodotorula glutinis* and rhamnolipids on cherry tomatoes infected with *Alternaria alternata*. They concluded that, even at low concentrations, the mixture of *R. glutinis* and rhamnolipids is a safe alternative for controlling *A. alternata* infection [9]. The screening of cultivation conditions with sophorolipids and the application of them at different stages of plant growth was investigated. In response, it was observed that sophorolipids present efficient biocontrol activity for biotic and abiotic stress in the primary stage of plant germination [10]. The production of the biosurfactant with an anionic characteristic from *Candida sphaerica* UCP 0995 was investigated, and then, the germination index was used to evaluate the toxicity of the biosurfactant in the germination of *Lactuca sativa* L., indicating that the solutions of 0.125, 0.25, and 0.5 g/L did not inhibit the germination of seeds or the elongation of roots [11]. The biosurfactant production from *Candida lipolytica* UCP 0988 at 0.15, 0.3, and 0.6 g/L did not inhibit the germination of the seeds of *Lactuca sativa* L. [12].

MELs are glycolipid biosurfactants [13,14]. The acetylation-based classification of MELs includes MEL-A, MEL-B, MEL-C, and MEL-D. In this sense, MEL-B has an acetyl group in its chemical structure [15–17]. The application of biostimulants has been reported in agricultural practices; however, there is no practical research on the biostimulant activity of MELs in seed cultivation.

However, the surface-active properties of biosurfactants exhibit relevant pesticidal and antimicrobial properties [18]. Thus, biosurfactants are a promising alternative that may lead to the sustainable management of pathogens and agricultural pests, partially reducing chemical pesticides and contributing to a more sustainable agricultural practice [19,20]. The hydrophilic and hydrophobic portions of biosurfactants contribute to promoting interactions between immiscible liquids in agricultural pesticide formulations. Therefore, understanding biosurfactants, biostimulants, and biopesticide behavior expands opportunities in the surfactant market [19,21].

The continued use of chemical pesticides in agricultural practices has had a negative impact on ecosystem health and also on plant development. This major environmental problem can be tackled through environmentally correct solutions using the properties of biosurfactants to totally or partially reduce plant pathogenicity and increase the concentration of chemical pesticides in the environment [19,22].

Thus, among biosurfactants, MELs are well-reported in the literature for presenting promising results on their antimicrobial activity against pathogens associated with food and crop management [23].

Therefore, from the correlation between the properties of MELs and other glycolipids with potential application in agriculture, a screening of the concentration of MEL-B was carried out to evaluate the biostimulant activity of MEL-B in seeds of Monica lettuce (*Lactuca sativa* L.), taking into account the morphological behavior, physiological characteristics, and physical-chemical and biochemical analyses performed after the germination phase.

## 2. Material and Methods

### 2.1. Material

Monica lettuce (*Lactuca sativa* L.) seeds were purchased from the company Feltrin Sementes Ltd.a.(Farroupilha/Brazil); Sgima and Merck (Florianópolis/Brazil), registered in the National Registry of Seeds and Seedlings (RENASEM) of the Brazilian Ministry of Agriculture, Livestock and Supply (MAPA). MEL-B at 95% purity was kindly provided by TOYOBO CO., LTD., Osaka, Japan. Guaiacol, sodium phosphate, phosphoric acid, hydrogen peroxide, PBS, and BSA reagents of analytical grades were obtained from Sigma and Merck.

2.1.1. Growing Medium Containing MEL-B for Lettuce Seed Germination

The germination tests were carried out in a Petri dish containing purified agar (% agar) at different concentrations of MEL-B (0, 3.16, 31.6, 158, 316, and 632 mg/L). Simultaneously, the MEL-B was weighed on an analytical balance (AD-500, Marte, São Paulo/Brazil) and subsequently solubilized into agar using a vortex mixer (K45-2820, Kasvi, São Paulo/Brazil). After homogenizing, the media were transferred to Petri dishes inside a flow chamber. The plates were sealed and stored in the refrigerator until use [7].

2.1.2. Contact Angle and Surface Tension

The influence of different concentrations of MEL-B (0, 3.16, 31.6, 158, 316, and 632 mg/L) on the measurement of contact angle and surface tension was performed in a goniometer where drops of soybean oil or diiodomethane were placed on the surface of agar containing MEL-B by using a micropipette of 100 μL. The procedure was performed in triplicate at 25 °C. The drops were photographed by a digital camera (Ramè-Hart, 250-F1, São Paulo/Brazil). The photo was subjected to digital processing to obtain the width and height [24]. The Drop Image provided the contact angle and surface tension values.

*2.2. Germination Test*

Lettuce seeds were sterilized with an aqueous alcohol solution (95% alcohol) for 5 min, and then the seeds were subjected to a hypochlorite solution (2%) for 1 min and posteriorly abundantly washed with distilled water. Then, the 100 seeds were distributed in 10 plates and incubated in a BOD chamber (New Lab, NL-41-02, São Paulo/Brazil) with controlled relative humidity (60%) for seven days, with day and night simulation at 25 and 20 °C, respectively. The number of lettuce germinated seeds was monitored for each concentration of treatment with MEL-B (0, 3.16, 31.6, 158, 316, and 632 mg/L). The germination speed index (GSI) and mean germination time (MGT) were calculated using Equations (1) and (2) [25].

2.2.1. Germination Speed Index (GSI)

The germination speed index of emerged seedlings was carried out on a daily basis and calculated according to Maguire [26]:

$$GSI = {N1}/{D1} + {N2}/{D2} + \cdots + {Nn}/{Dn} \tag{1}$$

in which *GSI* = germination speed index; $N1$, $N2$, $Ni$ = number of seeds germinated in the first count, second count, i-th count, respectively; $D1$, $D2$, $Di$ = number of days in the first count, second count, i-th count, respectively. Unit: dimensionless.

2.2.2. Mean Germination Time (MGT)

The plates containing lettuce seeds were monitored daily, and the average germination time was calculated as proposed by Labouriau [27]:

$$MGT = {\sum ni \times ti}/{\sum ni} \tag{2}$$

in which *MGT* = mean germination time; $ni$ = number of seeds germinated in time $ti$ (not the accumulated number but the one referred to the *i*-th observation); $ti$ = time between the beginning of the experiment and the *i*-th observation. Unit: days.

*2.3. Morphological Parameters in Lettuce Cultivation*

The seeds were cultivated with different concentrations of MEL-B. The behavior of lateral roots, stressed roots, length, and mass were evaluated. Regarding the appearance of lateral roots, the emergence of lateral roots was evaluated from the 3rd day of cultivation. One hundred Petri dishes containing 10 seeds per dish were monitored. All samples were observed visually and under a magnifying microscope (Technical, stereoscopic) [28].

Stressed roots were counted, as they did not germinate and/or showed low performance. In addition, the seeds that showed delay in the germination process during the observation of the 7 days of the experiment were taken into account. The accumulated records were represented in percentage at the end of the experiment.

The length of the roots (cm) was measured from the third day of germination to the seventh day. The procedure was performed in triplicate [29].

The mass of the samples used to prepare the crude enzymatic extract was determined using an analytical balance (Marte, AD-500). Seedlings (leaves and roots) were collected from Petri dishes daily and weighed. For each concentration, three runs were performed.

Morphological Analysis by Scanning Electron Microscopy (SEM)

The lettuce root samples were fixed with glutaraldehyde (2.5%) for 30 min. Then, they were dehydrated with an alcohol series (10, 30, 50, 70, 80, 90, and 100%) and dried at room temperature. For the analysis, the lettuce was distributed on carbon tapes on the surface of stubs and then coated with a layer of gold. After recovering, the samples were analyzed in SEM (JEOL JSM (6390LV)), with a tungsten electron source secondary electron detector at 10 kv [30].

*2.4. Physicochemical Characterizations of Total Proteins and Activity of Peroxidase and Polyphenol Oxidase Enzymes*

The study of the influence of MEL-B on germination and the induction of stress conditions in cultivation was carried out by quantifying total proteins and analyzing the activity of peroxidase and polyphenol oxidase enzymes.

Crude Enzymatic Extraction

Crude enzyme extraction was performed daily from the 3rd day of cultivation. After this period, the seeds that visibly started the germination process were selected. To extract the enzymes, the selected roots were weighed and macerated in a crucible under an ice bath. The addition of 1 mL of 50 mM sodium phosphate buffer was added until a homogeneous mass was obtained. Then, the plant material was transferred to microcentrifuge tubes and then centrifuged at $15,952 \times g$-force for 10 min. The supernatant was used to determine the enzymes activity and protein content [31].

*2.5. Protein Content*

For protein quantification, the Lowry method was used [32]. A total of 100 µL of crude extract and 2 mL of solution C ($Na_2CO_3$ (2%)) in 1M NaOH and $CuSO_4$ (0.5%) were added to the test tubes. The mixtures stand for 10 min. Subsequently, 200 µL of Folin reagent was added, homogenized, and left to rest for another 30 min. The reading was performed under absorbance at a wavelength of 750 nm and calculated concerning the mass (g) of the sample used to prepare the crude extract. Mean of the absorbances obtained was obtained to determine the protein concentration in each analyzed sample. The calibration factor of the calibration curve was determined and, finally, the protein concentration for each sample was estimated according to the equation:

$$C = \frac{Abs \times F}{m} \tag{3}$$

in which $C$ = concentration of protein in each sample, $F$ = calibration curve factor, $Abs$ = Absorbance of sample, and $m$ = mass of the sample (g).

This determination was performed in triplicate.

2.5.1. Peroxidase Activity

A 140 µL aliquot of sodium phosphate buffer (50 mM, pH 6.4) containing 0.3% ($v/v$) guaiacol was used. An aliquot of 100 µL of crude enzyme extract and 60 µL of $H_2O_2$ (0.3%) was added. Enzyme activity was determined by spectrophotometer by observing the variation in absorbance at 470 nm and 25 °C for 5 min [33].

2.5.2. Polyphenoloxidase Activity

Polyphenoloxidase (PPO) activity was performed according to the methodology presented by Matsuno and Uritani [34]. This analysis was determined using catechol (0.02 mol/L) as a substrate for the enzyme. The reading was determined in proportions of 0.30 mL of sample and 1.85 mL of 0.10 M solution of phosphate buffer pH 6.0 with catechol. The absorbance was read in a UV-Vis spectrophotometer (Spectra Max, 384 plus) at 395 nm. The reading was performed every 1 min for 10 min, and water was used as a blank.

*2.6. Statistical Analysis*

The experimental plots consisted of 100 seeds at each concentration of treatment with MEL-B for daily monitoring of germination and 1.600 destructive samples to evaluate the behavior of germinated seeds and physical-chemical analyses, totaling 2.200 analyzed seeds in the period of 7 days of cultivation. Data were submitted to a one-way analysis of variance (ANOVA) significance test, and the difference was compared using Tukey's test ($p \leq 0.05$).

**3. Results**

*3.1. Contact Angle and Surface Tension*

Figure 1 shows the influence of different concentrations of MEL-B on the interaction of drops of diiodomethane (DIIM) and soybean oil with the treated surfaces.

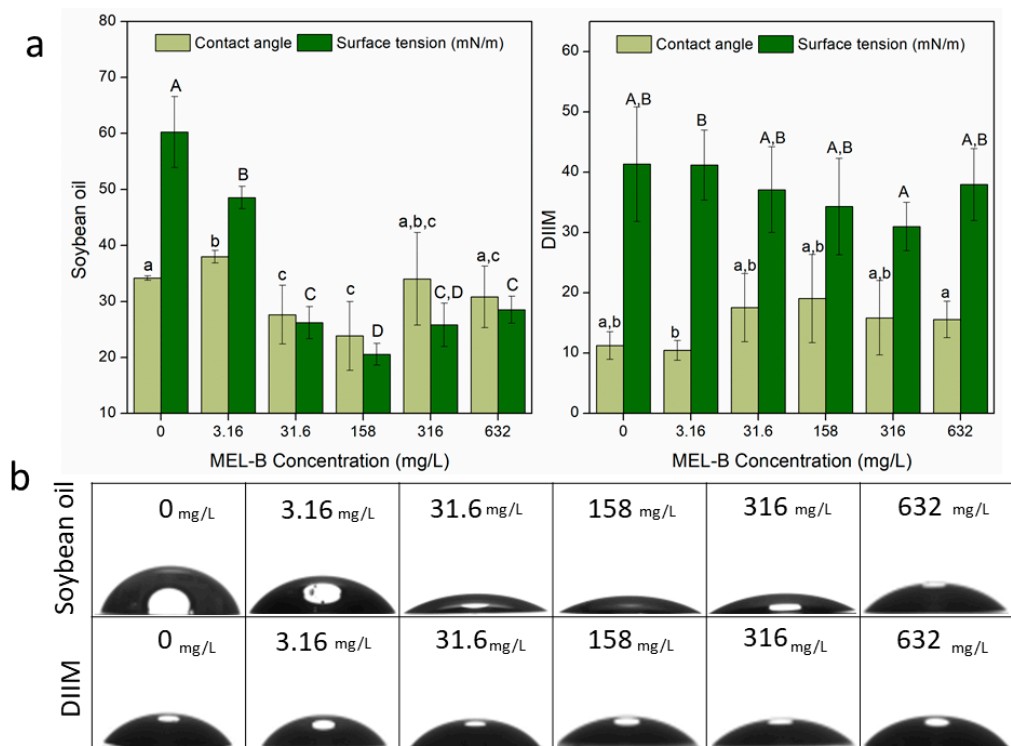

**Figure 1.** Effect of the drops of DIIM and soybean oil in the treated surfaces. (**a**) Contact angle and surface tension using oil and diiodomethane, (**b**) oil and diiodomethane drops in contact with the surface of the medium containing different concentrations of MEL-B. Means followed by the same letter do not differ from each other by the Tukey test ($p \leq 0.05$).

Figure 1a shows that this molecule did not undergo a significant variation in contact angle and surface tension after contact with agar surfaces treated with different concentrations of MEL-B. Therefore, despite having a contact angle lower than 90° and showing a wetting aspect, MEL-B did not promote the interaction of DIIM at the surface. Furthermore, the variation of MEL-B concentrations did not imply the reduction of the contact angle, as shown in Figure 1b.

On the other hand, a different response was observed after adding soybean oil. The surface tension reduced as the MEL-B concentration increased up to 31.6 mg/L, followed by a surface tension (quasi)plateau for higher MEL-B concentrations. MEL-B at 158 mg/L showed a significant difference in surface tension reduction about the control (Figure 1a). The behavior of the contact angle corroborates with this speculation since from the treatment carried out with 31.6 mg/L of MEL-B, wettability tended to increase. Due to the glycolipidic characteristics of MEL-B, the culture media supplemented with the biosurfactant suffered a weakening of the binding of water molecules.

### 3.2. Germination Properties

Figure 2 shows the influence of the MEL-B concentration added to the media on the number of germinated lettuce seeds (Figure 2a), germination speed index—GSI and mean germination time—MGT (Figure 2b), and the appreciation of secondary roots appearance (Figure 2d).

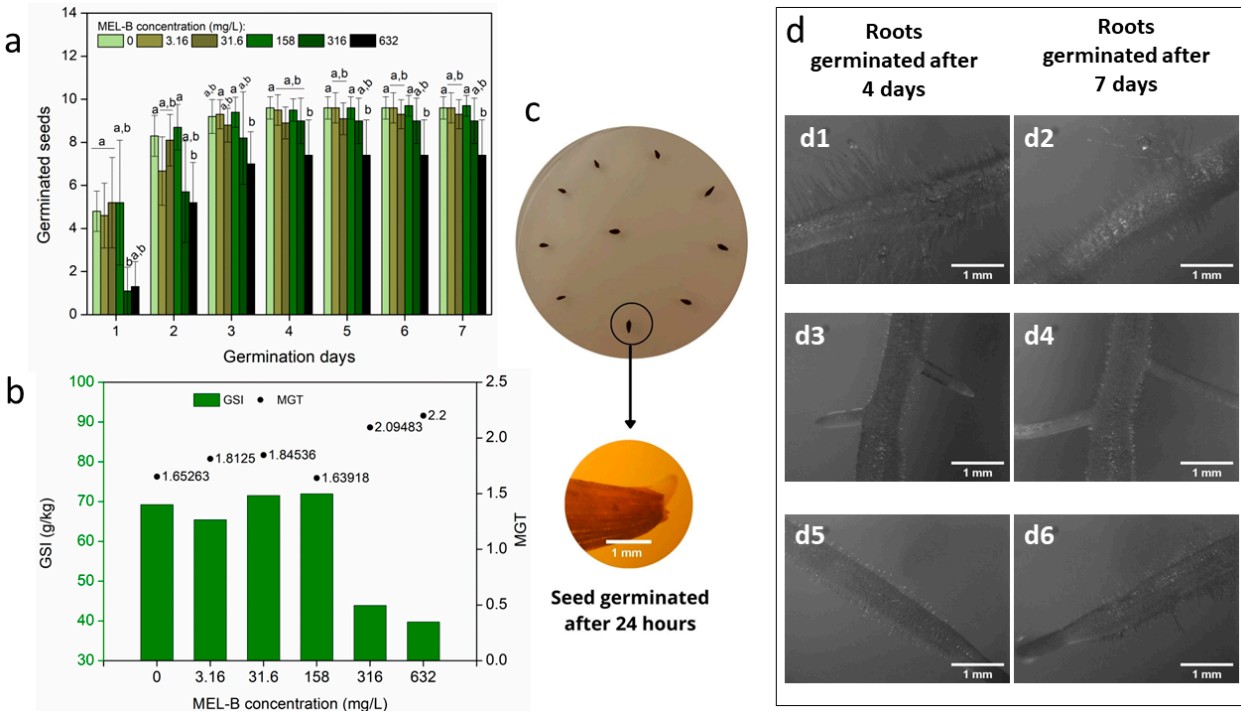

**Figure 2.** Effect of the different concentrations with MEL-B on the germination of *Lactuca sativa* L. (**a**) Cumulative germination of seeds, (**b**) GSI represented by vertical bars and MGT represented by points, and (**c**) representative image of lettuce seeds germinated after 24 h of cultivation. (**d**) Optical microscopical image of lettuce seeds germinated after 4 and 7 days of cultivation. Means followed by the same letter do not differ from each other by the Tukey test ($p \leq 0.05$).

Figure 2c reports the characteristic behavior of seeds that germinated after 24 h of cultivation. The observation was realized using an optical microscopical at $10\times$ enlargement. The concentrations of MEL-B added to the culture medium influenced the incidence of germination (Figure 2a); however, all the seeds germinated on the 1st day showed the same morphological characteristics, independent of the MEL-B content present.

The incidence of germination was observed cumulatively during the germination of lettuce seeds (Figure 2a). On the 1st day of the experiment, it was observed that MEL-B at 316 and 632 mg/L affected seed germination, where only about 15% of the seeds germinated out of 100%. In contrast, at lower concentrations (0, 3.16, 31.6, and 158 mg/L) of MEL-B, more seeds (greater than 40%) germinated under each treatment condition. The differences in relation to the control were significant ($p \leq 0.05$) only in the concentration of 316 mg/L of MEL-B; it was noticed that the seeds cultivated with 316 and 632 mg/L of MEL-B

germinated less in comparison with the other conditions. The germination was higher than 80% in all growing conditions, except for the concentration of 632 mg/L of MEL-B, which was the highest used for seed germination and had lower levels since the first day of germination (Figure 2a).

One of the indicators of seed vigor is the GSI, which is directly proportional to each other. That is, the higher the GSI, the more vigorous the seed [34]. Regarding GSI, MEL-B promoted similar responses for control and intermediate conditions (3.16, 31.6, and 158 mg/L), indicating values above 65% for GSI. Differing significantly from concentrations of 316 and 632 mg/L, GSI showed an inhibitory effect by MEL-B with values of 43.9 and 39.7%, respectively (Figure 2b).

The results obtained for the GSI were corroborated by the MGT, where the time required for germination was greater for concentrations of 316 and 632 mg/L of MEL-B. For the culture containing 158 mg/L of MEL-B, there was a decrease in MGT compared to subsequent concentrations, returning to an increase in following treatments. This observation demonstrates that the average germination time of lettuce seeds is progressively increased under biotic stress. Treatment with MEL-B reduced GSI, increasing MGT at 316 and 632 mg/L. At the concentration of 158 mg/L, the results for the same parameters were the opposite, confirming a less pronounced inhibitory effect than the other cultivation conditions (Figure 2b).

Figure 2c shows the behavior of all roots in the first 24 h of cultivation. In all conditions, the same behavior of the germinated seeds was observed. Figure 2d shows morphological observations of the germinated roots after 4 and 7 days of cultivation. Figure 2 panels d1 and d2 demonstrates the predominant behavior of seeds grown in the medium without MEL-B treatment (control) in the medium containing MEL-B at 3.16 and 31.6 mg/L. In the records represented on Figure 2 panels d3 and d4, the evolutionary behavior of the germinated seeds in the culture medium containing MEL-B at 158 mg/L was compiled. The treatment performed with MEL-B at 316 and 632 mg/L is represented in Figure 2 panels d5 and d6. Thus, observations recorded with the aid of a microscope indicate that the different treatments with MEL-B caused morphological changes in the roots.

### 3.3. Morphology of the Roots

Root growth was monitored from the 4th day of seed germination (Figure 3). It was noted that on this 1st day of observation, all cultivation conditions showed similar behavior. All verified roots showed similar sizes in the measurements (~1.8 cm).

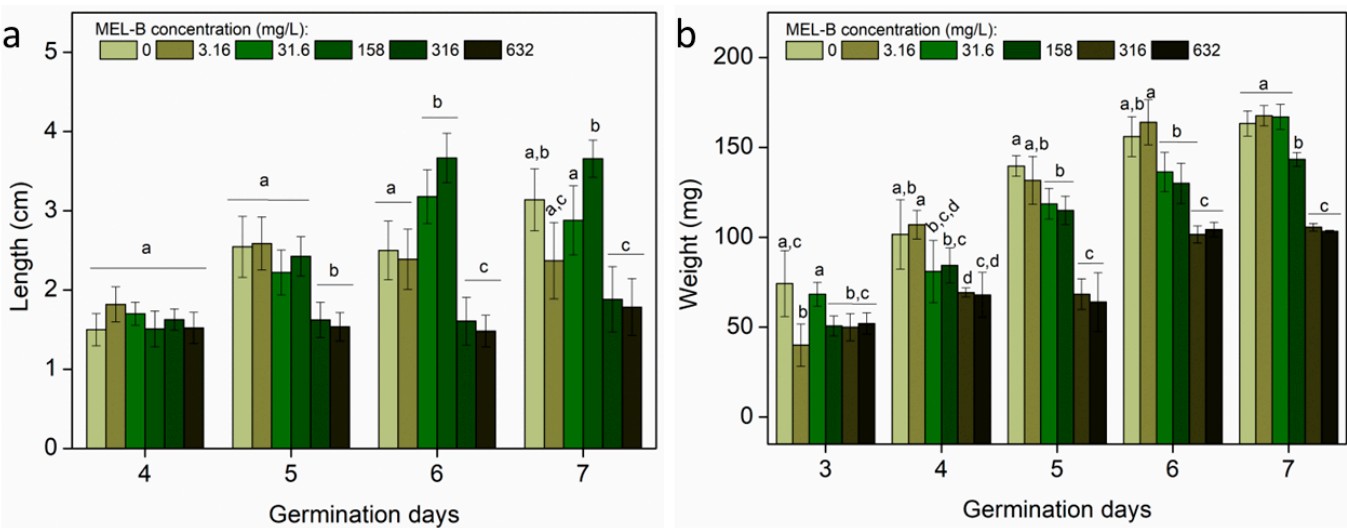

**Figure 3.** Monitoring of different concentrations of MEL-B on the morphology of *Lactuca sativa* L. roots. (**a**) Length and (**b**) weight of roots. Means followed by the same letter do not differ from each other by Tukey's test ($p \leq 0.05$).

However, from the 5th day of monitoring, the concentrations of 316 and 632 mg/L of MEL-B showed a significant difference in relation to the size of roots grown under control conditions. On the 6th day of cultivation, the roots cultivated at a concentration of 158 mg/L of MEL-B reached the largest size (less than 3.5 cm in length) compared to the other treatment conditions. In addition to presenting the largest size among the concentrations, it was the largest size observed among all the experiment days. In general, the seeds grown without MEL-B and with low concentrations (3.16, 31.6, and 158 mg/L) of MEL-B tended to grow with the days of cultivation. However, seeds germinated with 316 and 632 mg/L of MEL-B were less than 2 cm in length and did not show significant development since the 1st day of monitoring.

Another parameter monitored was the weight of the germinated roots (Figure 3b). On the 3rd day of the experiment, the roots germinated in the control conditions showed an average difference of 34.33 mg in relation to the roots germinated with medium containing 3.16 mg/L of MEL-B. On the 4th day of germination, the weights of the roots grown in the control and 3.16 mg/L of MEL-B were >100 mg and showed a significant difference in treatments made with 316 and 632 mg/L of MEL-B, which were lower than only 70 mg. From the 5th day, the behavior was similar for all concentrations. The tendency to increase in weight as the roots developed was observed. However, at MEL-B concentrations of 316 and 632 mg/L, the same pattern observed in Figure 3a was observed for the same treatment conditions. The roots developed less when compared to the control and other concentrations. On the last day of monitoring, the roots that developed the most (in this parameter) were without treatment and with 3.16 and 31.6 mg/L of MEL-B, obtaining an average weight >160 mg. The opposite was observed for roots treated with 632 mg/L of MEL-B. The lowest average weight (<105 mg) was observed in this condition compared to the control and the other concentrations on the 7th day.

Factors that cause adverse reactions in the development of lettuce seeds were noted in the experiment (Figure 4). The treatment performed with MEL-B on seed germination promoted the development of lateral roots (Figure 4a) at intermediate concentrations and created a stress (Figure 4b) medium at higher concentrations in addition to the morphological observations, as illustrated in Figure 4c.

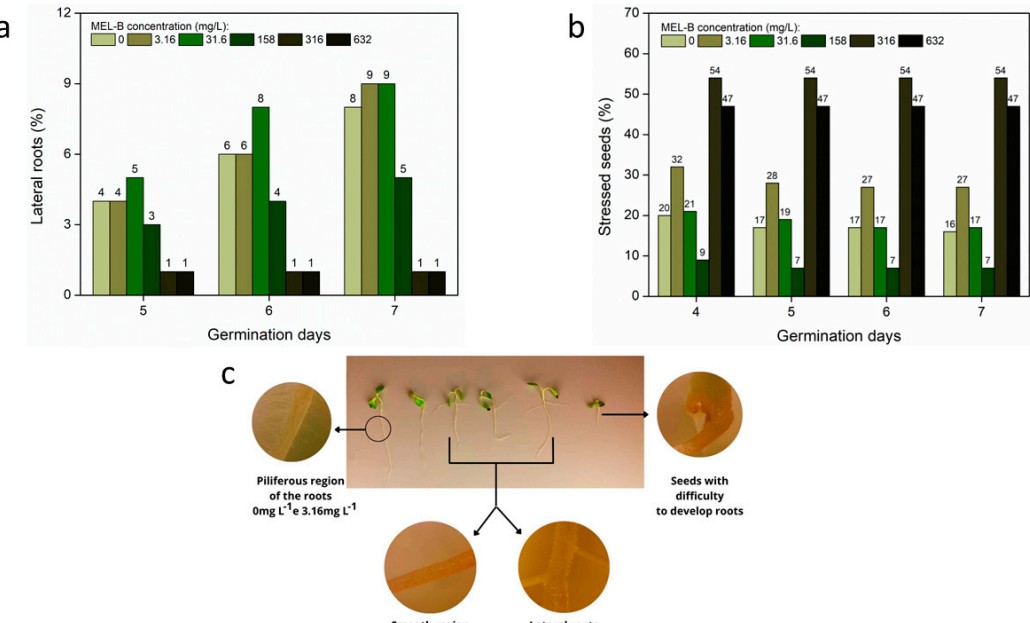

**Figure 4.** Monitoring (**a**) lateral roots, (**b**) stressed roots, and (**c**) illustrative image of root regions on the seventh day of cultivation under different concentrations of MEL-B.

As observed on the 4th day of germination (Figure 4b), there were increases of 34 and 27% for treatments made with 316 and 632 mg/L of MEL-B compared to the control. For

the treatment with MEL-B at 158 mg/L, the stressed roots were reduced by 11% when compared to the control. From the 5th day of cultivation, intermediate concentrations (3.16, 31.6, and 158 mg/L) and the control showed reduced stressed seeds. This behavior may reflect seed dormancy, a phenomenon that causes an intrinsic temporal block that provides additional time for germination. On the other hand, seeds grown under treatments of 316 and 632 mg/L of MEL-B remained with 54 and 47% of the stressed samples (Figure 4b). At the end of the experiment, it was noticed that the treatment performed with 158 mg/L of MEL-B presented only 7% of the roots stressed. That was the only concentration that showed a reduction of stressed roots among the treatments made with MEL-B compared to the control.

However, the treatment performed with MEL-B at 31.6 mg/L stimulated the same number of lateral roots in seven days of cultivation (Figure 4a). This behavior was also observed when seeds were germinated with MEL-B at 316 and 632 mg/L (Figure 4b).

The influence of different treatments with MEL-B is also recorded in Figure 4c. The behavior of the germinated roots after 7 days of cultivation shows that the intermediate treatments (3.16, 31.6, and 158 mg/L of MEL-B) did not inhibit the development of the roots. The opposite was registered when the seeds were germinated in the culture medium treated with MEL-B at 316 and 632 mg/L. Under these conditions, the roots showed adverse behavior in relation to the control and other treatments.

The microstructural analysis of the root surface was performed after 7th days (Figure 5). SEM is another alternative that makes it possible to interpret the surface of the roots, evaluate the microstructure, and correlate the potential influences of treatments with MEL-B in the cultivation. The evaluated roots presented plant tissue with an irregular shape and contracted cellular aspect in all treatments performed. In this way, the cell walls characterized the appearance of withered plant cells.

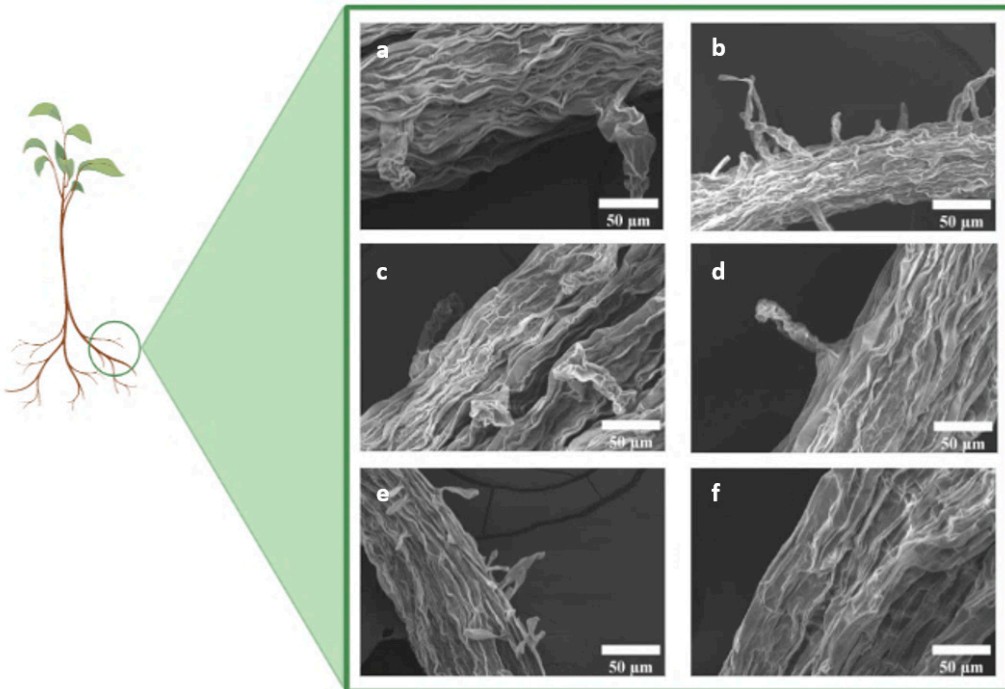

**Figure 5.** Scanning electron microscopy (SEM) of radicle and primary root of Monica lettuce after seven days of cultivation. (**a**) Root grown without treatment, and Radicle treated with MEL-B (**b**) 3.16 mg/L, (**c**) 31.6 mg/L, (**d**) 158 mg/L, (**e**) 316 mg/L and (**f**) 632 mg/L.

Under control conditions and treatment with 3.16 mg/L of MEL-B, the plant tissue of the primary roots showed a well-developed hairy region (Figure 5a,b). Although apparently in smaller amounts, the development of the piliferous region was observed in seeds cultivated under treatment of 31.6 and 158 mg/L of MEL-B (Figure 5c,d). For 7 days

of cultivation, the seeds treated with up to 158 mg/L of MEL-B presented root growth superior to those treated with 316 and 632 mg/L of MEL-B. The observations made with SEM corroborate the behavior illustrated in Figure 4c, where a visual comparison of root development under different treatment concentrations was performed.

Furthermore, it was noted that the plant tissue of the primary roots of seeds treated with 316 mg/L of MEL-B had a shape and structure similar to those observed in the root tissue of seeds germinated with 632 mg/L (Figure 5f). However, a considerable reduction in the development of the hairy region with these treatments was seen. In addition, the integrity of the roots was compromised. However, with a low rate of seeds germinating under these conditions, the root growth and morphological development were significantly lower than the control and other treatment concentrations.

### 3.4. Quantification of Protein and Enzyme Activity

The protein quantified in each treatment condition was evaluated from the crude extract of the sprouted roots. The crude extract obtained by the roots on the 3rd day of germination was produced with roots of lower size and mass than the roots that germinated for 4 days and successively until the 7th day of germination. In this sense, it was observed that even though the root mass decreased with increasing concentrations of MEL-B in the culture medium (Figure 3b), the protein quantification remained at similar values and without a statistical difference ($p \leq 0.05$)—for the roots germinated until the 4th day of the experiment (Figure 6a). This physiological alteration can corroborate with the interpretation of the behavior of the germinated roots under treatment of 316 and 632 mg/L of MEL-B and validate the stress that these concentrations cause in the germination of the seeds. From the 4th day of the experiment, it was noted that all concentrations presented a higher or equal amount of protein than the control. In addition, the highest amount of protein obtained in this study was 11.33 mg/g on the 5th day of germination, with the extract of germinated roots under treatment performed with 632 mg/L of MEL-B.

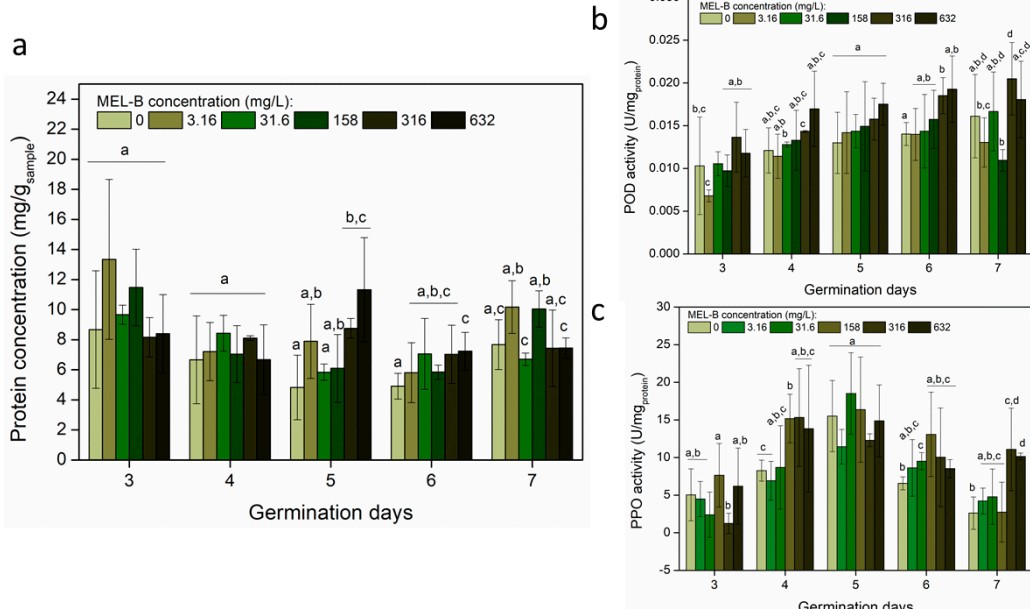

**Figure 6.** Effect of different concentrations of MEL-B on the physiology of *Lactuca sativa* L. roots. (**a**) Total protein in cultivated lettuce seeds, (**b**) Peroxidase enzymatic activity, and (**c**) Polyphenoloxidase. Means followed by the same letter do not differ from each other by Tukey's test ($p \leq 0.05$).

Peroxidase and polyphenol oxidase enzymes are pathogenesis-related enzymes involved in the cell wall lignification process and plant defense development processes in response to biotic and abiotic stresses.

When analyzing the peroxidase activity (Figure 6b), MEL-B treatments caused changes in the peroxidase enzyme activity. On the 3rd day of germination, only concentrations of 316 and 632 mg/L of MEL-B showed higher enzyme activity than the control. In the evaluations carried out from the fourth day onwards, it was noted that there was no difference in enzyme activity between the treatments and the control. On the 7th day of the experiment, the highest enzyme activity was 0.0205 $U/mg_{protein}$ in the treatment with 316 mg/L of MEL-B. On the other hand, the concentration of 158 mg/L showed the lowest enzyme activity for the same day of culture. In general, regarding enzymatic activities, the results showed that the levels of the peroxidase enzyme were low. However, plants may have suffered oxidative stress after treatment with MEL-B, which resulted in higher levels of peroxidase enzyme activity at higher treatment concentrations.

The polyphenol oxidase enzyme (Figure 6c) had its activity increased until the 5th day of germination. On the 6th and 7th days, lower levels of activity were observed. The treatment performed with 31.6 mg/L of MEL-B showed 18.5 $U/mg_{protein}$ on the 5th day of germination, which was the highest level of enzyme activity among all the days of cultivation. On the 7th day of germination, treatments made with 316 and 632 mg/L of MEL-B showed the greatest differences in enzyme activity from the control at 8.48 and 7.54 $U/mg_{protein}$, respectively. In addition, treatments performed with 316 and 632 mg/L of MEL-B influenced lower root development when compared to control and intermediate concentrations. However, it was noted that the enzyme activity levels remained above 10 $U/mg_{protein}$. The increase in polyphenol oxidase activity occurred without increasing the protein concentration, indicating that the roots were subjected to water stress.

## 4. Discussion

### 4.1. Interpretation of the Behavior of the Contact Angle and Surface Tension

The hydration of the plant cell is essential for biochemical reactions and seed metabolism [35]. In this sense, MEL-B is an surface active agent—it reduces the surface tension [17]. DIIM is a highly non-polar molecule that is not very water-soluble [36]. Therefore, this behavior was expected since naturally, the DIIM molecule has a low surface tension [37]. Thus, the surface tension reduced as the concentration of MEL-B increased. In the case of the soybean oil drop, the observed behavior may be related to the lipophilic character of the biosurfactant [38].

In general, the behavior observed with soybean oil under different cultivation media can predict the behavior of the lettuce seeds in the present study. In this sense, the lower surface tension—due to the increase in MEL-B concentration in the culture medium—may reflect higher levels of seed wettability in contact with the treated surfaces. Therefore, greater seed hydration is expected along with, consequently, a more significant influence of the bioactive properties of MEL-B on seed germination as biosurfactant concentrations increase.

### 4.2. Assessing Germination Properties

In observations of the germination incidence, the same incidence level in soybean seed germination (less than 80%) after treatment with rhamnolipids was observed. However, the concentrations of rhamnolipids used for seed treatment were higher than the concentrations used in this study [39]. Another study evaluated the effect of rhamnolipid on lettuce germination and growth. They noted that the concentration of 750 mg/L stimulated lettuce seed germination but impaired radicle development compared to control [29]. According to Karthika et al. [40], the biosurfactant produced by the *Bacillus* sp. also indicated an improvement in the germination percentage of tomato seeds after treatment.

This work observed that MEL-B showed more significant interaction with the external tissue surrounding the seed due to its chemical structure. MEL-B increased the permeability of the seeds and contributed to a better performance in the germination of the seeds in some concentrations [41].

### 4.3. Morphological Changes of Roots

The seeds subjected to rhamnolipid treatment in another study showed decreased root length as the concentration increased. This observation may suggest a phytotoxic effect of this by-product against seedlings at high concentrations. In relation to root mass, the highest treatment concentration with rhamnolipid (1 g/L) showed a lower mass than control and other treatments [29]. This behavior was also observed in the treatments with MEL-B at 316 and 632 mg/L in this work. In another study that used the biostimulant Coveron in lettuce germination, the observed effects were positive about the control, where the length of the lettuce roots grew up to 2.1 cm at the end of the experiment [42]. Compared to this study, MEL-B showed superior results, with roots up to 3 cm in length under treatment at a concentration of 158 mg/L.

Sophorolipids were applied to the germination of barley seeds. In 10 days of germination, 195 mg/L of this biosurfactant stimulated the development of nine lateral roots. In comparison to the control, the application of a sophorolipid was superior by 2% in the stimulation of the lateral roots [43]. Regarding stressed roots, researchers noted that the development of germinated roots after treatment with rhamnolipid at 1 g/L was lower compared to control and intermediate concentrations [29]. In this sense, these bioproducts have an inhibitory effect on seed germination at high concentrations. However, the biostimulant effect of MEL-B can be noticed in lower amounts when compared to rhamnolipids and sophorolipids.

Khare and Arora applied biosurfactants to *Lycopersicon esculentum* and showed antiphytopathogenic and biocontrol activities. Similar to MEL-B, the authors noted that the biosurfactant promoted increased root growth and improved plant evolution and potential antimicrobial activity in several spectrums and was also environmentally better than chemical pesticides [44].

On the 10th day of tomato seed germination, the authors reported that the vermicompost treatment promoted the growth of 3.57 cm of roots [45]. However, MEL-B obtained the same result with only 6 days of germination.

Thus, it can be said that the mechanisms of action of MEL-B in lettuce seeds contribute to the oxidative stress of roots when treated with MEL-B at 316 and 632 mg/L. Furthermore, this imbalance in plant tissue and root development indicates a lack of control in primary plant metabolism in which ATP synthesis may be compromised.

### 4.4. Biochemical Analyzes after Treatment with MEL-B

The reduction in germination and root length can be attributed to the decrease in cell divisions due to morphological and physiological changes caused by the treatment used in seed germination. Figure 6 illustrates the biochemical responses in relation to enzymatic activities and the quantification of total proteins after seed cultivation with MEL-B. Studies indicate that proteins are covalently linked to the lignin molecule and, therefore, associated with cellulose in the cell wall and can confer rigidity, impermeability, and resistance against biological attacks to plant tissues. In addition, the lignification of root tissues can promote anatomical changes and influence water absorption, affecting root cell elongation [46].

Regarding the evaluated enzymes, the size and mass of the roots decreased as MEL-B concentrations were increased. This fact may be related to the fact that the enzymatic activity was evaluated using the crude extract of these roots. The responses obtained in this step corroborate the observations made in the previous steps in this work, indicating that the highest treatment concentrations (MEL-B at 316 and 632 mg/L) caused stress in seed cultivation.

As the concentration of vermicompost increased sharply in seed germination, the authors noticed an increase in protein content, POD, and PPO activity [45]. The same behavior was observed for treatments performed with MEL-B at 316 and 632 mg/L.

## 5. Conclusions

Unprecedentedly, this is the first report on the influence of MEL-B on seed germination. MEL-B at 158 mg/L showed promising results in the biostimulation of cultivated seeds. On the other hand, the responses observed in the physiological and biochemical behavior indicate that MEL-B at 316 and 632 mg/L influenced oxidative stress and inhibited the germination and development of the seeds. However, it is fundamental to identify the mechanisms of biosurfactant-plant interaction. These biomolecules have great potential to replace chemical pesticides based on new formulations with biosurfactants, and the analysis of obtained results indicated that MEL-B has great potential to replace, even if partially, the chemical components present in conventional pesticides, aiming to combat phytopathogens and promote the application of MELs to improve the solubility and/or degradation of chemical pesticides, the biostimulation of plants, and the use of MELs to promote soil quality by removing heavy metals and crude oil.

**Author Contributions:** R.D.M., C.J.d.A., D.d.O., B.A.M.C. and K.C. developed the methodology; R.D.M. performed the experimental part and text writing; R.D.M., C.J.d.A., D.d.O., B.A.M.C. and K.C. performed data analysis; C.J.d.A., D.d.O. and B.A.M.C. critically reviewed the manuscript and language. All authors have read and agreed to the published version of the manuscript.

**Funding:** CAPES-PRINT: Project numbers 88887.310560/2018-00 and 88887.310373/2018-00 and CNPq (Conselho Nacional de Desenvolvimento Científico e Tecnológico) and CAPES-PROEX (Coordenação de Aperfeiçoamento Pessoal de Nível Superior Programa de Excelência Acadêmica).

**Institutional Review Board Statement:** This article does not contain any studies with human participants or animals performed by any of the authors.

**Informed Consent Statement:** Not applicable.

**Data Availability Statement:** The authors declare that the data supporting the findings of this study are available within the article.

**Acknowledgments:** The authors are grateful to Federal University of Santa Catarina for the infrastructure for the development of the work.

**Conflicts of Interest:** The authors declare that they have no conflict of interest.

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
