# Peer review of "The Biosurfactants Mannosylerythritol Lipids (MELs) as Stimulant on the Germination of Lactuca sativa L."

_agriculture, doi:10.3390/agriculture13091646_

Round 1

Reviewer 1 Report

THE BIOSURFACTANT MANNOSYLERYTHRITOL LIPIDS 2 (MELs) AS STIMULANT ON THE GERMINATION OF LAC-3 TUCA SATIVA L.

Rewrite the abstract. Briefly write the problem, methodology, results in quantitative terms and recommendation from the study. Delete line 23-26.

L-18: MEL, write the full name when it first appears. Delete “a promising alternative to modern agriculture”.

L-20: What is MEL-B and SF 31? B is coding given by you or some other acronym created by the authors.

L26-27: stimulated the germination…how much?? In comparison to other treatments, wheat was the performance of MLE-B? Likewise, also present the other results in per cent or quantitative form.

Please revise the introduction section too. Though the work is novel but novelty and research gaps have not been highlighted. Ideally the structure of the introduction part should be the major problem (which the authors have presented satisfactorily), followed by: the rest parts like present level of knowledge, research gaps, objectives, hypothesis etc., which needs to be improved significantly.

It is good that you wish to investigate the function of glycolipids called mannosyl erythritol lipids (MELs) as biosurfactants. However, I was unable to discern from the manuscript the significance of MEL, its application in agriculture, and the rationale for its selection as a biosurfactant.

Methodology part is okay. But, have you repeated the experiments? What is the probability of the repeatability of your results in the pragmatic on-farm environments? Generally, multi-season, or multi-location results are appreciated for field applicability studies.

In Figure 2 The author claims that the development of roots takes 24 hours.  There is not a picture of the seed in its growing stage in the manuscript. Regarding the development phase or callus formation with different intervals of time, no image is provided. Only a picture of root development is provided.

Mention some future agricultural, environmental, and bioremediation possible uses of your research in conclusion.

Minor improvements in English language will be fine

Author Response

  1. The abstract portion is quite good and informative. However, I will recommend to the author, please add a few sentences at the start of the abstract as introduction and study gaps, which leads to the objectives.

Thank you, the abstract was changed, significantly.

  1. In the abstract, the author directly mentioned MELs, which confuses the audience, so it’s better to use the full term in the abstract and then abbreviations.

It was corrected.

  1. The introduction needs to improve with the latest references. At this stage, the introduction is not enough, and we try to strengthen more with MELs studies, which is very important nowadays. Moreover, finds the gaps which lead to the author objectives.

We appreciate the suggestions and have added more up dated references. We also indicated the research gaps with the objective of the work.

  1. Material and Methods: How much agar was used for seeds germination per litter, how much and placed in the growth chamber for how many days or weeks (line 88)

that were added contributed to carry out new comparisons on the results.

MEL-B concentrations per liter of agar were added. The incubation period of the plates in the BOD chamber and the seed germination evaluation time are described in topic 2.2 Germination test.

  1. Material and Methods: The author washed the seeds with only ethanol, and why not bleaches (sodium hypochlorite) this time? As per recommendations, bleaches is a first step kind of.

We appreciate the observation made. 2% hypochlorite was used for 2 minutes after disinfection with alcohol, then the seeds were washed with plenty of water, as described in the methodology. This information has been added to the text.

  1. Results are almost fine; no need to change as per my knowledge.

Done.

  1. Some grammar and references should be updated in the discussion to give more strength.

We added new references, and to strengthen the discussion of the results. The enghish grammar was evaluated by professional-services.

  1. Overall I am satisfied with the discussion's startup, but I need to add the latest references and make better comparisons.

We appreciate the suggestions and have added more up dated references, including the discussion item.

Best wishes

Reviewer 2 Report

The bio-surfactant mannosylerythritol lipids (mels) as stimulant on the germination of lac-tuca sativa l.

Dear Editor,

The authors have provided important information on the bio-surfactant mannosyl-erythritol lipids (mels) as a stimulant on the germination of Lactuca productivity and its growth development. Agronomic and ecological advantages can also be expected from this kind of work in organic farming which would help enhance the audience knowledge and minimize the study gaps regarding and protection of soil and plant health in bio-organic agriculture. The manuscript is organized and informative. However, I will give some suggestions for improving the manuscript regarding scientific language and errors. I hope the author will improve it and then accept it.

General Comments

Ø  In the overall manuscript, grammatical corrections are suggested. Furthermore, references need to be double check and updated per journal policy, and some new references should be added in the introduction.  

Comments:

1.      The abstract portion is quite good and informative. However, I will recommend to the author, please add a few sentences at the start of the abstract as introduction and study gaps, which leads to the objectives.

2.      In the abstract, the author directly mentioned MELs, which confuses the audience, so it’s better to use the full term in the abstract and then abbreviations.   

3.      The introduction needs to improve with the latest references. At this stage, the introduction is not enough, and we try to strengthen more with MELs studies, which is very important nowadays. Moreover, finds the gaps which lead to the author objectives.

4.      Material and Methods: How much agar was used for seeds germination per litter, how much and placed in the growth chamber for how many days or weeks (line 88)

5.      Material and Methods: The author washed the seeds with only ethanol, and why not bleaches (sodium hypochlorite) this time? As per recommendations, bleaches is a first step kind of.

6.      Results are almost fine; no need to change as per my knowledge.

7.      Some grammar and references should be updated in the discussion to give more strength.

8.      Overall I am satisfied with the discussion's startup, but I need to add the latest references and make better comparisons.

Best wishes 

The bio-surfactant mannosylerythritol lipids (mels) as stimulant on the germination of lac-tuca sativa l.

Dear Editor,

The authors have provided important information on the bio-surfactant mannosyl-erythritol lipids (mels) as a stimulant on the germination of Lactuca productivity and its growth development. Agronomic and ecological advantages can also be expected from this kind of work in organic farming which would help enhance the audience knowledge and minimize the study gaps regarding and protection of soil and plant health in bio-organic agriculture. The manuscript is organized and informative. However, I will give some suggestions for improving the manuscript regarding scientific language and errors. I hope the author will improve it and then accept it.

General Comments

Ø  In the overall manuscript, grammatical corrections are suggested. Furthermore, references need to be double check and updated per journal policy, and some new references should be added in the introduction.  

Comments:

1.      The abstract portion is quite good and informative. However, I will recommend to the author, please add a few sentences at the start of the abstract as introduction and study gaps, which leads to the objectives.

2.      In the abstract, the author directly mentioned MELs, which confuses the audience, so it’s better to use the full term in the abstract and then abbreviations.   

3.      The introduction needs to improve with the latest references. At this stage, the introduction is not enough, and we try to strengthen more with MELs studies, which is very important nowadays. Moreover, finds the gaps which lead to the author objectives.

4.      Material and Methods: How much agar was used for seeds germination per litter, how much and placed in the growth chamber for how many days or weeks (line 88)

5.      Material and Methods: The author washed the seeds with only ethanol, and why not bleaches (sodium hypochlorite) this time? As per recommendations, bleaches is a first step kind of.

6.      Results are almost fine; no need to change as per my knowledge.

7.      Some grammar and references should be updated in the discussion to give more strength.

8.      Overall I am satisfied with the discussion's startup, but I need to add the latest references and make better comparisons.

Best wishes 

Author Response

Rewrite the abstract. Briefly write the problem, methodology, results in quantitative terms and recommendation from the study. Delete line 23-26.

Thank you, the abstract was changed, significantly.

L-18: MEL, write the full name when it first appears. Delete “a promising alternative to modern agriculture”.

It was corrected.

L-20: What is MEL-B and SF 31? B is coding given by you or some other acronym created by the authors.

The chemical structure of these molecules varies in the position and number of the acetyl radical. In this sense, MELs can be classified according to 4 different structures (MEL-A, MEL-B, MEL-C ,and MEL-D).

SF 31 is a reference code of the brand where the group obtained the seeds. However, they were removed from the text since it does not present relevant information for conducting the experiments.

L26-27: stimulated the germination…how much?? In comparison to other treatments, wheat was the performance of MLE-B? Likewise, also present the other results in per cent or quantitative form.

Changes and comparisons were made as suggested.

Please revise the introduction section too. Though the work is novel but novelty and research gaps have not been highlighted. Ideally the structure of the introduction part should be the major problem (which the authors have presented satisfactorily), followed by: the rest parts like present level of knowledge, research gaps, objectives, hypothesis etc., which needs to be improved significantly.

We appreciate the suggestions and have added more current references. We also listed the research gaps with the objective of the work.

It is good that you wish to investigate the function of glycolipids called mannosyl erythritol lipids (MELs) as biosurfactants. However, I was unable to discern from the manuscript the significance of MEL, its application in agriculture, and the rationale for its selection as a biosurfactant.

We have up dated the references, and changed the text in order to make more clear the application and selection of the MEL-B to conduct this investigation.

Methodology part is okay. But, have you repeated the experiments? What is the probability of the repeatability of your results in the pragmatic on-farm environments? Generally, multi-season, or multi-location results are appreciated for field applicability studies.

The entire experimental part was performed in triplicate to verify the methodology's reproducibility and the results' reliability. The investigated conditions were the optimal conditions for monica lettuce cultivation. It is noteworthy that due to the experimental repetitions, there was optimization in the methodological process. However, it is part of the objective of this study to evaluate the biostimulant property of MEL-B from direct application in the cultivation soil (agar) since similar results with MEL-B were not found in the literature. Thus, this study becomes relevant because it creates another field of investigation, which is studies with MEL-B applications in the field, being able to evaluate different variables, such as location, season, cultivation time, and water deficit, among others.

In Figure 2 The author claims that the development of roots takes 24 hours.  There is not a picture of the seed in its growing stage in the manuscript. Regarding the development phase or callus formation with different intervals of time, no image is provided. Only a picture of root development is provided.

Dear reviewer, thank you for looking at the figure, however records of all germination days were not provided for all conditions of treatment with MEL-B since the purpose of the figure is to represent the relevant changes during the analyzed period. In short, the changes physiological changes observed in seed roots. Thus, figure 2c represents the evolution of seed germination in the first 24 hours observed, and Figure 2d illustrates the behavioral pattern of the roots from different treatment concentrations – what makes it relevant for the discussion are the changes promoted in the structure of the roots, such as: established hairy region and disappearance of this region for certain treatments and also the stimulation and inhibition of secondary roots in the germination period.

Mention some future agricultural, environmental, and bioremediation possible uses of your research in conclusion.

We added some possible applications of MEL-B for the improvement of agricultural practices, listing the use of MELs in the fight against phytopathogens, the application of MELs to improve the solubility and/or degradation of chemical pesticides, and the use of MELs to promote soil quality from the removal of heavy metals and crude oil.

Reviewer 3 Report

In line 103 add hours to a day/night regime

The main objective of this study was to evaluate the biostimulatory effect of MEL -B on 69 seeds of Monica SF 31 lettuce (Lactuca sativa L.). Key aspects such as seed germination kinetics, vegetative propagation, modulation of root architecture, and attenuation of root stress responses were investigated.

- The problem has been studied from different aspects, which gives this manuscript a particular value.

- The manuscript is clearly and comprehensively written.

- The results obtained are well compared with the existing literature.

- Most of the references cited are recent publications.

-The figures and tables are appropriate and show the results correctly.

The article is well-written and structured. It falls within the journal's scope as it is a valuable topic.

The methods used in the article are adequately described and explained in a way that is easy to understand for a wide range of readers. The processed results are well-displayed and carefully structured. The listed references are relevant to the study. 

Author Response

Dear, thank you for take your time in order to provide valuable comments on the manuscript